# Towards Combinatorial Generalization for Catalysts: A Kohn-Sham Charge-Density Approach

**Phillip Pope**
University of Maryland, College Park
pepope@cs.umd.edu

**David Jacobs**
University of Maryland, College Park
dwj@umd.edu

## Abstract

The Kohn-Sham equations underlie many important applications such as the discovery of new catalysts. Recent machine learning work on catalyst modeling has focused on prediction of the energy, but has so far not yet demonstrated significant out-of-distribution generalization. Here we investigate another approach based on the pointwise learning of the Kohn-Sham charge-density. On a new dataset of bulk catalysts with charge densities, we show density models can generalize to new structures with combinations of elements not seen at train time, a form of combinatorial generalization. We show that over $80\%$ of binary and ternary test cases achieve faster convergence than standard baselines in Density Functional Theory, amounting to an average reduction of $13\%$ in the number of iterations required to reach convergence, which may be of independent interest. Our results suggest that density learning is a viable alternative, trading greater inference costs for a step towards combinatorial generalization, a key property for applications.

## 1   Introduction

The Kohn-Sham (KS) equations are a nonlinear eigenvalue problem of the form $H[\rho]\Psi = E\Psi$, where $H$ is a symmetric diagonally dominant matrix called the *Hamiltonian*, $\Psi$ is an eigenvector, $E$ the associated eigenvalue, and $\rho(\mathbf{r}) = \sum_i |\Psi_i(\mathbf{r})|^2$ is a real-valued scalar field called the *charge density*, which is unknown a priori [40, 30]. The KS equations are nonlinear in the sense that the matrix $H$ depends on the charge density $\rho$, which in turn depends on the eigenvectors $\Psi$ of $H$. Typically it is solved by fixed-point iteration, where an initial guess for $\rho$ is made and then a sequence of *linear* eigenvalue problems are solved until convergence. In the computational chemistry literature this is referred to as the *self consistent field* (SCF) iteration. The cost of the iteration is dominated by the eigenvalue problem [41]. Consequently methods of reducing the number of requisite iterations, e.g. with machine learning, are of great interest.

The KS equations lie at the foundation of Density Functional Theory (DFT), an approach to electronic structure theory which reformulates the $N$-particle quantum many-body problem in terms of an effective one-particle density [31]. DFT is widely used in a number of physico-chemical applications. One such application with important economic and environmental consequences and growing attention from the machine-learning community in recent years is the discovery of new catalysts [54].

Perhaps the greatest difficulty of catalyst discovery is the combinatorial size of the search space of structures. Even when reducing the candidate set of elements to 55, as done in the Open Catalyst Project [8], the number of possible element combinations grows very quickly: we have $\binom{55}{3} = 26,235$, $\binom{55}{4} = 341,055$, and so forth. Additional factors like choice of crystal lattice, surface orientations, binding sites, and adsorbates further complicate matters, but nevertheless the number of possible element combinations is a dominating factor. Given the immensity of this search space, *generalization to new combinations* is a key aspect for the success of property-predicting models in applications. Put

37th Conference on Neural Information Processing Systems (NeurIPS 2023).

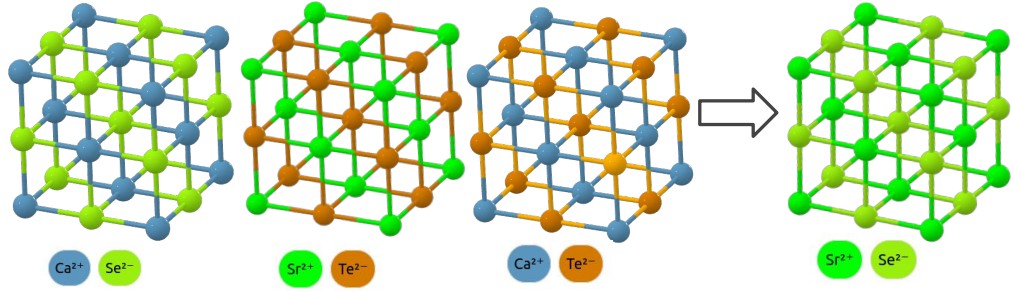

Figure 1: Simple illustration of a combinatorial generalization task with binary catalysts $\{CaSe, SeTe, CaTe\}$ for training and $\{SrSe\}$ for testing. Can density values of the structures on the left be used to predict the density values on the right? Note the combination of elements occurring in the test set does not occur in the training set, but its constituent elements do. Materials project IDs from left to right are `mp-1415`, `mp-1958`, `mp-1519`, and `mp-2758`. Structures visualized with the Materials Project web app [24]. NB: density values are not shown.

differently, predictive models that cannot combinatorially generalize will fail to cover large parts of the search space. Moreover some authors argue that combinatorial generalization is an intrinsically important property for machine learning models in general [4].

Recent research in machine-learning for catalyst modeling aims at directly predicting the energy and/or per-atom forces on large-scale benchmarks [8, 48]. The energy is a global property of a physical system which, among other reasons of theoretical importance, helps to assess the efficiency of catalytic reactions [54]. This line of work has spawned a number of innovations in equivariant and geometric graph learning [6, 55, 12, 11]. Despite these advances, generalization performance as measured by the Energy within Threshold (EWT) metric has not achieved greater than $20\%$ of examples in any test split at the time of this writing [1]. It remains unclear if further development or scaling will lead to practical results.

An emerging alternative to energy prediction in the context of DFT systems is the *pointwise* prediction of the charge density [53, 17, 5, 27, 39]. The charge density is a local rather than global property of the system: a density value is associated to each point in the computational domain. The density is fundamental in DFT in the sense that all other properties may be computed from it, e.g. the energy by way of the eigenvalue problem, potentially eliminating the need for property-specific models in favor of a single model.

In this work we carry out an empirical investigation of density based models in catalyst systems. The focus of our investigation is whether such models can generalize to new combinations of elements. Our contributions are as follows:

- We compute a new large-scale DFT dataset of $\mathcal{O}(1000)$ relaxed bulk catalyst structures *with charge densities* spanning $\mathcal{O}(100M)$ unique points using the open-source DFT software Quantum Espresso and workflow software AiiDA [37, 15, 21, 22, 49, 38].

- We design an evaluation methodology to measure when learned densities improve convergence versus standard baselines for density initialization in DFT. To do so we define a threshold-independent metric and also report the number of iterations saved at convergence.

- We show that density models improve convergence on new structures not seen at train time. Specifically we find that learned densities outperform baselines in $83\%$ and $86\%$ of test cases for binary and ternary catalyst respectively. These savings amount to a reduction of $13\%$ in the number of SCF iterations needed to reach convergence.

To the best of our knowledge, our results are the first density-based approach to show improvement over standard baselines for density initialization in DFT on new structures not seen at train time.

## 2 Related Work

### 2.1 Catalyst Modeling

The discovery of new catalysts has received significant attention from the machine-learning community in recent years. Recent work in this area has primarily been lead by the Open Catalyst Project (OCP) [8]. Building on a line of works in geometric graph learning for atomic systems [51, 42], OCP has contributed a number of modeling developments [55, 12, 11, 46, 13] and large-scale datasets [8, 48]. See Joshi et al. [26] for a summary of recent approaches based on the level of geometric information incorporated in the model.

### 2.2 Learning the Kohn-Sham Density

A notable advantage of density-based approaches is that it naturally interfaces with the DFT algorithm, something not possible with energy models. Density predictions may be directly passed to the SCF cycle, which made be used to (1) better initialize a new SCF cycle versus standard baselines or (2) adaptively refine the predictions to increase the precision. In contrast the outputs of energy models are fixed and cannot be refined in this way.

Perhaps the most similar work to ours is that of Gong et al. [17], who use a graph-neural-network based approach for density prediction which encodes the local environment with a query node as we do. Using the CGGCNN [51] model, a predecessor to more advanced equivariant and geometric graph models that we use here, they perform a small-scale study of model transferability to new kinds of bonds not seen at train time, e.g. train on linear C-C-C and orthogonal C-O-C and test on orthogonal C-C-C (See Figure 7 of [17]).

Another similar work on density prediction is that of Zepeda-Núñez et al. [53], who represent an atomic structure as a *set* rather than a graph and define a novel network architecture with translation, rotation and permutation symmetries. They show up to four-digits of accuracy on non-catalyst test structures, including water, small organic molecules, and aluminum.

Other notable works include (1) Rackers et al. [39] who use Euclidean neural networks [14] to study generalization from smaller to larger clusters of water; (2) Brockherde et al. [5] who use a kernel-based approach to learn the dynamics of small organic molecules, Grisafi et al. [18] study transfer learning of density models from small molecules to larger ones such as hydrocarbons, (3) Jørgensen and Bhowmik [27], who show small-scale results with a message-passing network using "two-step approach" to encode the structure graph and local environment separately, in contrast to the query node approach that Gong et al. [17] and the present work uses, and (4) Schütt et al. [43] who learn elements of the Hamiltonian matrix directly for single molecule cases using a modified Schnet model [42], and show savings in SCF iterations in select cases.

### 2.3 Combinatorial Generalization

Combinatorial generalization (CG) is generalization from simpler data to more complex data [4]. CG and related ideas have been the subject of many machine learning works across several domains including generative models [34, 23, 33], vision and language [52, 47], reinforcement-learning [25, 7, 50], visual and analogical reasoning [20], and puzzle-solving [3].

To the best of our knowledge there are few priors works that explicitly address CG in chemistry-related machine learning. Fernando [10] explores related ideas in the context of chemical systems. Gui et al. [19] propose an out-of-distribution graph learning benchmark with molecular tasks but do not explicitly focus on any combinatorial structure as we do here.

### 2.4 High-precision learning

Most modern machine learning models trained through stochastic optimization do not typically achieve training loss values around machine-precision, $10^{-16}$ for double-precision `float64`. One

notable exception is Michaud et al. [32], wherein the authors demonstrate a training method enriched with the spectrum of the Hessian for achieving near machine-precision loss performance on low-dimensional examples such as $y = x^2$. It is unclear if such techniques scale to larger models . We note the distinction between high-precision at train time, a problem of optimization, and high-precision at test time, a problem of *generalization*. Colbrook et al. [9] argue there are fundamental limits to the precision obtainable by neural networks.

## 3 Methods

### 3.1 Modifying geometric graph learning for local property prediction

Standard geometric graph learning typically predicts a global property of a structure/molecule, e.g. the energy or band gap. Here we modify this approach for local property prediction by adding a "virtual atom" representing the query point to the graph. This approach is similar to Gong et al. [17], except our case deals with geometric graphs.

Let $s = \{(\mathbf{R}_i, Z_i)\}_{i=1}^I$ denote a structure $s$ with $N$ atoms at positions $\mathbf{R}_i \in \mathbb{R}^3$ with atomic numbers $Z_i$. Let $\rho_s : \mathbb{R}^3 \to \mathbb{R}$ denote the density function associated to a structure $s$, evaluated on a grid of query points $\{\mathbf{r}_j\}_{j=1}^J$, with all $\mathbf{r}_j \in \mathbb{R}^3$.

In the usual formulation, radial graphs are typically created from each structure, i.e. draw an edge between two atoms if they lie within some radius of each other. To instead make the graph local to a point, we first adjoin the query point to the structure as a "virtual" atom $(\mathbf{r}_j, Z_{I+1})$, where we interpret the atomic number $Z_{I+1}$ simply as just an unused index on atom types. Note we use the same index for all query points. Letting $\mathcal{G}(\cdot)$ denote the operation of taking the radial graph, the localized graph then takes the form $\mathcal{G}(s \cup (\mathbf{r}_j, Z_{I+1}))$. Denote all such pairs for a structure as $\mathcal{D}_s = \{\mathcal{G}(s \cup (\mathbf{r}_j, Z_{I+1})), \rho_s(\mathbf{r}_j)\}$. Lastly, we complete the dataset by taking a union over all structures $\mathcal{D} = \bigcup_s \mathcal{D}_s$.

Given this data, we proceed as usual to stochastically optimize parameters $\theta$ of a graph neural network GNN to minimize the expected loss $\mathcal{L}$ of samples $g, \rho$ drawn from a (uniform) probability measure $\mathbb{P}_\mathcal{D}$ on the dataset $\mathcal{D}$:

$$\min_\theta \mathbb{E}_{g,\rho \sim \mathbb{P}_\mathcal{D}} \left[ \mathcal{L}\left(\text{GNN}_\theta(g), \rho\right) \right] \tag{1}$$

Since densities are continuous real-valued quantities, a regression loss is used.

### 3.2 Creating a dataset of catalyst charge densities with Quantum Espresso and AiiDA

Although there are emerging datasets of charge-densities [45], our evaluation requires that the settings of the data and solver be *exactly the same* to ensure numerical consistency. To that end, we generate our own dataset of relaxed bulk catalysts using the well-established and open-source DFT solver Quantum Espresso [16, 15] and modern computational workflow management system AiiDA [21]. In particular we utilize the relaxation workflows of Huber et al. [22], which automate the selection of computational parameters critical for achieving convergence based on expert knowledge.

For initial candidates for bulk catalysts we use the list of bulk catalysts identified by [8] from the Materials Project [24], which contains $11,410$ structures sampled from 1, 2 or 3 element combinations of a set of 55 elements. The total number of element combinations represented in this set is 5099.

Grid sizes per structures can vary by up to four orders of magnitude, which can strongly skew the training data toward larger structures. To mitigate this we restrict the maximum number of atoms to be 12 and the maximum volume of the (conventional) cells to $200 \, \text{Å}^3$. We then remove candidates with oxidation states which only occur once in the dataset, since these are not well-represented enough to evaluate, using the oxidation state guesser in `pymatgen` [35].

We then partition the train/val/test splits as follows. First we assign all unary catalysts remaining to the train set. Then we randomly sample binaries with oxidation states not already represented in the unaries. Next, we remove oxidation states not present in the training set from the remaining binary and ternary candidates. This preprocessing step performed to ensure all states in the val/test splits

Table 1: Sample sizes of dataset

|                | $N_{\text{structures}}$ | $N_{\text{samples}}$ |
|----------------|-------------------------|----------------------|
| Train (unary)  | 47                      | 3.8M                 |
| Train (binary) | 392                     | 69M                  |
| Val (binary)   | 4                       | 300k                 |
| Test (binary)  | 360                     | 72M                  |
| Test (ternary) | 1116                    | 380M                 |

were represented in the training set, e.g. training on $H^+$ cannot be expected to generalize to $H^-$. Finally we assign a few binaries to the validation split and the remaining binaries and ternaries to the testing split. Finally we validate that train and test splits are disjoint by element combinations, i.e. no combination of elements in training set is repeated in the test splits.

We relax each structure in these splits using Quantum Espresso with parameters set by the AiiDA relaxation workflow [22]. We use the `fast` protocol with the PBE exchange-correlation functional [36] for all relaxations. Initial structure positions were first perturbed with noise. Unconverged runs were dropped from the dataset. We report the number of structures and density samples in Table 1.

### 3.3 Evaluating convergence of learned densities versus standard baselines in DFT

An important baseline for density based models is the "atomic charge superposition" (ACS) initialization for charge densities, commonly used in DFT solvers [2]. The ACS baseline is derived from the pre-computed collection of atomic *pseudopotentials* used with the DFT solver, an important technique that reduces the number of electrons in the computation. For further details see [41, 30, 31, 29].

More precisely, given a configuration of atoms $\{(\mathbf{R}_i, Z_i)\}_{i=1}^I$ and associated atomic charge densities $\rho_{Z_i}$, the ACS initialization is $\rho_0^{ACS}(\mathbf{r}) = \sum_i^I \rho_{Z_i}(\mathbf{r} - \mathbf{R}_i)$. In other words it is simply a sum of radial functions. Note this initialization corresponds to a scenario of non-interacting atoms: poor performance may be expected when the atoms interact, i.e. when there is chemistry.

In this work we consider learned densities that do not improve upon this baseline to *not* be of interest. From a practical point of view, clearly initializations which do not beat this baseline would not be relevant. Nevertheless, given the difficulty of obtaining high precision results with learning-based approaches, this baseline is highly non-trivial.

The degree of precision required depends on the application. One could set a threshold and check which density reached convergence in fewest iterations. However, we wish to perform the evaluations in a threshold independent way to remain agnostic to the requirements of application, similar in spirit to "area-under-the-curve" (AUC) measures popular in classification.

To that end we propose a "signed area under curves" (s-AUC) measure, essentially the area between the curves of convergence for the baseline and learned densities. See Figure 6 for examples. Note the sign is necessary because the curves may cross. More formally, if $\{x_i\}_{i=1}^{n_x}$ and $\{y_j\}_{j=1}^{n_m}$ are the SCF accuracy values for the baseline and learned curves respectively, and $n = \min(n_x, n_y)$ we define

$$\text{s-AUC} = \frac{\sum_{i=1}^n \log(x_i) - \log(y_i)}{\sum_{i=1}^n |\max(\log(x_i), \log(y_i))|} \tag{2}$$

where we have used a log-scale to show rate of convergence and normalized by the pointwise max to facilitate comparison between difference curves.

We also report the percentage of iterations saved at convergence. Although this measure is threshold dependent, it has the benefit of being easily interpretable to the ML+DFT community.

### 3.4 Training with Spherical Channel Networks

We train a reduced sized Spherical Channel Network (SCN) for density prediction. We reduce the number of interactions to 10 and hidden channels to 256, resulting in 35M parameters, which is

smaller than state of the art models [55]. We use interaction cutoff to 6.0Å, and a maximum number of neighbors per node of 12. We list full hyperparameters in the Supplementary.

We train two replicates of SCNs for 420k steps with batch size 45 across 8 GPUs. Regarding compute, for training we used approximately 2000 GPU hours for both replicates across all GPUs. We use `NVIDIA A5000` cards with 24 GB of memory.

In initial experiments we also evaluated Dimenet++ [11] and GemNet [13] models but generally found performance on these models to be less than SCN models.

## 4 Results

### 4.1 Model performance

In training we obtain mean absolute errors (MAE) of $\mathcal{O}(10^{-4})$. Best validation MAE was $0.0011$ for both replicates. One replicate was chosen to report the remainder the results. We show the distribution of MAE scores across each train/test split in Figure 2.

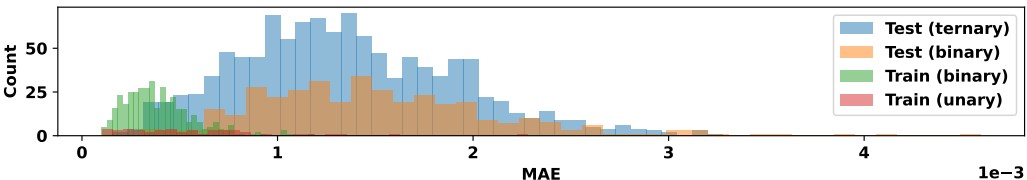

Figure 2: Distributions of MAE scores for train/test splits

### 4.2 Visualization of learned densities and errors

One advantage of local property prediction is the ability to directly visualize errors as a function of input points. In Figure 3 we show the true density, predicted density, and (scaled) absolute error for the top and bottom test predictions by MAE.

These visualizations allow us to inspect of the distribution of errors, which may facilitate e.g. model debugging, among other uses. Here we see a variety of patterns: ranging from nearly non-existent to quite strong errors near the core region of atoms. A closer analysis of these error patterns may inform future improvements to the dataset and model.

### 4.3 Evaluation of learned density convergence

We evaluate if initializing with learned densities leads to faster convergence in the SCF cycle. After predicting across the grid associated to each structure, we pass predictions to the DFT solver to initialize a new SCF cycle using *exactly the same runtime parameters* as the ground-truth run. Then we extract convergence information and compare them against the ground truth.

We show summary statistics of our results on SCF savings according to the s-AUC metric in Table 2. Notably we achieve positive savings in $83\%$ of test binary cases and $86\%$ of ternary cases, with a combined proportion of $86\%$. The mean and median savings in each case are positive.

In addition to the s-AUC metric, we also report savings in terms of the relative percentage of iterations saved at convergence in Table 3. The percentage of test structures with positive savings was $71\%$ and $75\%$ of binary and ternary test cases. The mean percentage of iterations saved was $13\%$ for both binary and ternary splits.

We show the distribution of s-AUC scores and iteration savings across splits in Figures 4 and 5.

W plot the convergence behavior of top test cases by s-AUC in Figure 6. Positive savings can be seen when blue-dashed curves lie beneath the red-solid lines.

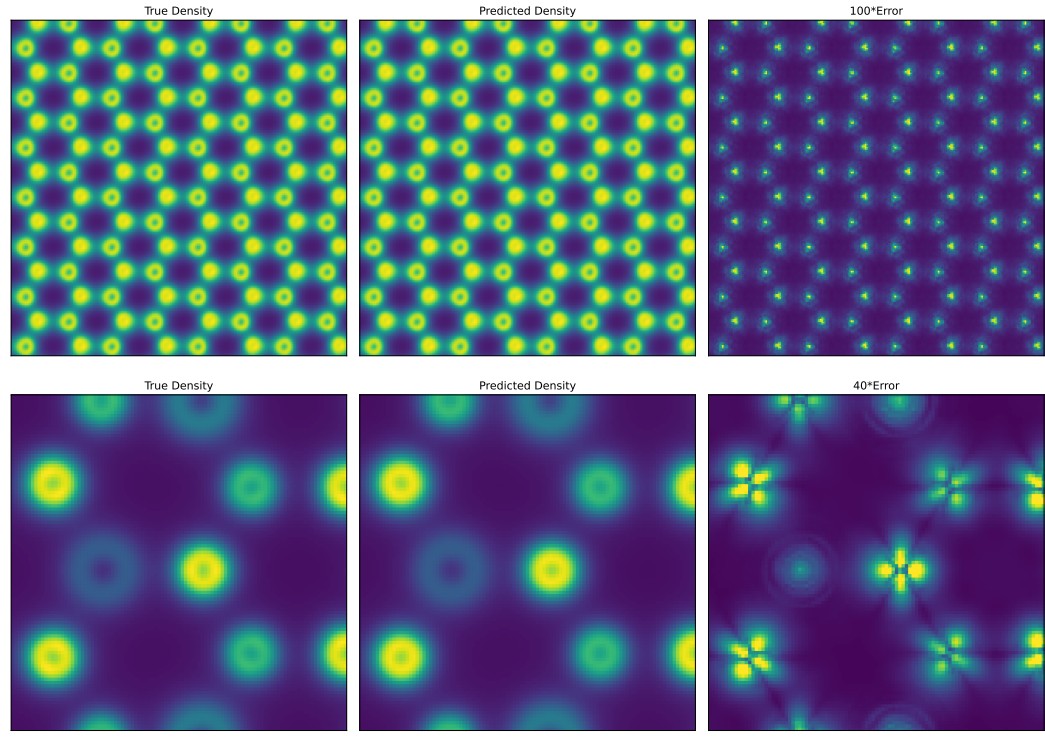

Figure 3: Test densities (2D projections): ground truth, predicted, and error. Top row is low error test structure $MoS_4W$(mp-1023954) and bottom row is higher error test structure $Fe_6Re_2$(mp-865212) with higher MAE error. Errors rescaled display more clearly. To visualize 3D densities are regridded to approximately a cube [44] and summed along one axis.

Table 2: Summary statistics of SCF savings with the s-AUC metric.

|                | N    | $N_+$ | $N_+/N$ (%) | Mean | Median | Max               | Min                  |
|----------------|------|-------|-------------|------|--------|-------------------|----------------------|
| Train (unary)  | 47   | 45    | 96          | 0.6  | 0.3    | 7.3               | -0.1                 |
| Train (binary) | 392  | 379   | 97          | 4.0  | 0.6    | $2.9 \times 10^2$ | -0.5                 |
| Test (binary)  | 360  | 298   | 83          | 7.8  | 0.2    | $1.9 \times 10^3$ | -3.7                 |
| Test (ternary) | 1116 | 965   | 86          | 2.7  | 0.4    | $7.2 \times 10^2$ | $-2.8 \times 10^2$   |

Table 3: Summary statistics of relative iterations saved at convergence.

|                | N    | $N_+$ | $N_+/N$ (%) | Mean (%) | Median (%) | Max (%) | Min (%) |
|----------------|------|-------|-------------|----------|------------|---------|---------|
| Train (unary)  | 47   | 32    | 68          | 15       | 14         | 78      | -53     |
| Train (binary) | 392  | 361   | 92          | 22       | 21         | 63      | -50     |
| Test (binary)  | 360  | 256   | 71          | 13       | 12         | 72      | -53     |
| Test (ternary) | 1116 | 841   | 75          | 13       | 12         | 65      | -90     |

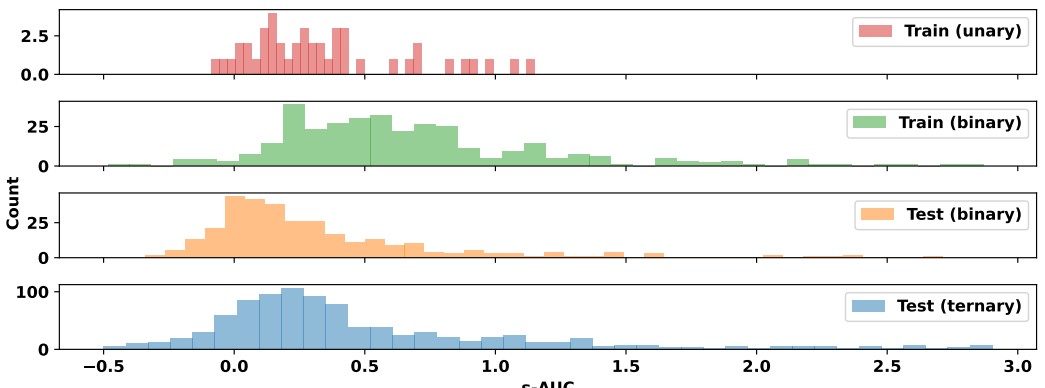

Figure 4: Distributions of `s-AUC` metrics. Outliers dropped for visualization.

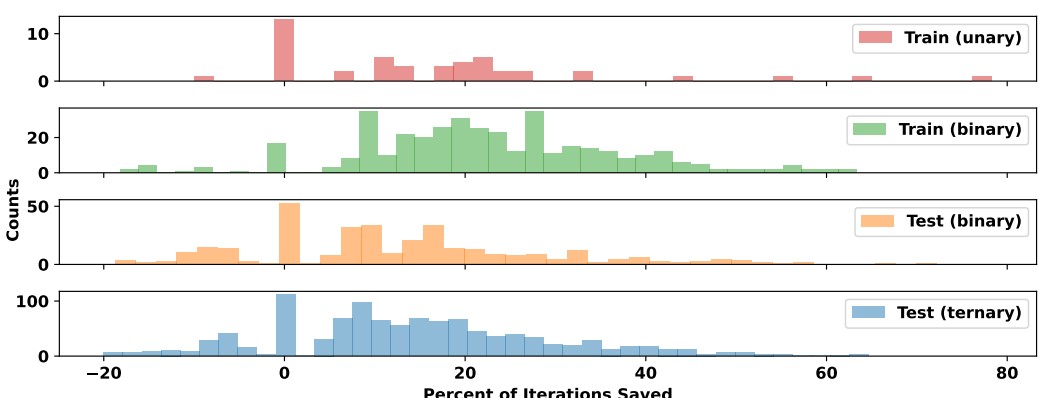

Figure 5: Distributions of relative iteration savings. Outliers dropped for visualization.

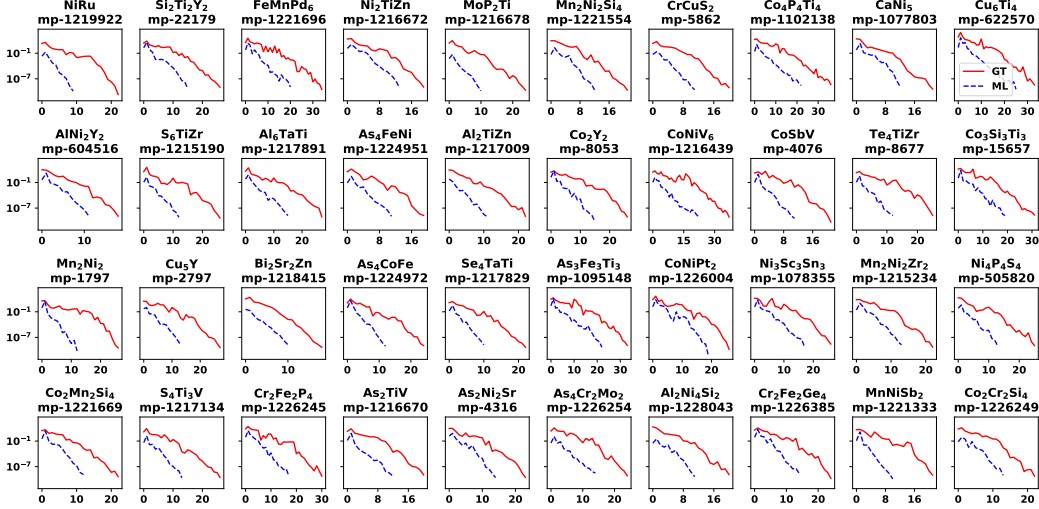

Figure 6: Top convergence results for test structures by s-AUC metric. Red solid lines are ground-truth DFT convergence, blue dashed lines are learned densities. Y-axis is SCF accuracy on a log-scale and x-axis is number of SCF iterations. Chemical formulas and material project ids given above each subplot. Note these structures were not seen at train time.

### 4.4 Validating learned densities

We perform two validations to check if learned densities converge to the same solution as the ground truth.

First we checked for any bias in converged energy values between ground-truth and learned densities. We find almost zero difference these energy values. On the test set we found $88\%$ of structures had exactly zero difference in energy values. Of the cases with non-zero error, the relative error on average was $1.2 \times 10^{-11}$ and at maximum was $1.5 \times 10^{-9}$.

Second, we check if a downstream property computed from the learned densities matches their ground-truth counterparts. We compute and compare the band gaps computed from the ground-truth and ML densities using the built-in tool in ASE [28]. We find $95\%$ of values to be exactly the same, with maximum relative error at $7.8 \times 10^{-5}$.

## 5 Discussion

Overall we find in over $80\%$ of test cases learned densities achieve faster convergence versus the baseline. Importantly these test cases are out-of-distribution, involving structures with new combinations of elements not seen at train time. Our results show that training on simpler structures, i.e. unary and binary, can lead to generalization on more complex structures, i.e. ternary.

## 6 Limitations

In this work we only test generalization for combinations of elements, and ignore stoichiometric ratios. In addition, we only investigate bulk catalysts, not the full complexity of catalyst systems as modeled in Chanussot et al. [8] which includes adsorbates, surfaces, and non-equilibrium points. Extending our analysis to these more complex systems is left to future work.

Our energy values are not directly comparable with those of Chanussot et al. [8], because we generated our dataset with a different DFT solver and pseudo-potentials. Systematic numerical differences may exist between different solvers, which makes comparison of energy values difficult.

Another limitation is that we do not report timings of our method versus traditional CPU-bound methods. Because of the high GPU-bound inference costs of our method, we do not expect timing improvement versus traditional methods. Sufficiently many GPUs will help mitigate this, as well as recent hardware development trends towards larger GPU memory.

## 7 Conclusion

We generated a new dataset of bulk catalyst relaxations with charge densities using open-source software. We trained a density-model pointwise on unary and binary catalysts. We showed this model can generalize to new binary and ternary catalysts, a form of combinatorial generalization. We found learned densities lead to faster convergence than standard baselines used to initialize the SCF cycle of DFT in over $80\%$ of cases, amounting to an average of $13\%$ saving in iterations.

### 7.1 Future Work

There are several directions for improvement to this work. The first is improving the proportion of test samples with positive savings, where clearly higher is better. In addition, expanding the class of structures tested, e.g. to quaternary structures and beyond, is of keen interest in applications.

A second direction is improving the rate of convergence of the learned densities. At present a number of SCF iterations are still necessary to reach high levels of convergence e.g. $10^{-9}$. The ideal case would be to converge rapidly to a single or few SCF iterations.

Rapid convergence across many kinds of structures opens up the interesting possibility of an end-to-end hybrid learning/numerical model which first precisely predicts the density then computes the energy in one or a few SCF iterations. Such models may have different Pareto curve properties than current energy-based approaches, and may be useful in applications with high precision requirements.

## 8 Broader Impact

One way this work may have positive broader impact is through reducing costs of DFT computations. For example, supposing that the USA National Labs spend $100M on DFT per year, and a negligible cost of inference, then a 13% reduction in SCF iterations would yield a savings of $13M [1]. Although there are a number of practical details to resolve, we believe this work is an important first step.

The potential impacts of new catalysts discoveries may have significant economic and environmental consequences. Positive impacts may include accelerating the transition towards renewable energy, an important goal for future societies. Negative impacts may include unintended consequences such as contributing to unsustainable population growth. In addition, as with any new technology with potentially large impacts in the economy, there is risk that control of the technology may concentrate in the hands of the few and limit the distribution of benefits.

In early stages of research is difficult to discern if the aims will come to fruition. The computational discovery of new catalysts does not necessarily imply that the results are translatable into real-world materials, e.g. not all materials are known how to be made easily or economically. If such technology can be discovered, we believe that the benefits can be more equitably distributed by making this research completely open-source and accessible to global researchers. Our use of open-source DFT solvers is one notable step towards this goal over previous approaches [8].

## 9 Acknowledgements

PP thanks Professors Maria Cameron, Howard Elman, Tom Goldstein, Ramani Duraiswami, Pratyush Tiwary, and Hong-Zhou Ye, as well as Dr. Ruiyu Wang for feedback on this project. PP also thanks the University of Maryland Institute for Advanced Computer Studies (UMIACS) for providing the computing infrastructure used in this project.

---

[1]We thank an anonymous reviewer for suggesting this way of interpreting our results

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
