# 1 Supplementary

Code and data to replicate our experiments can be found at https://github.com/ppope/rho-learn.

## 1.1 DFT Relaxations

We relax structures using Quantum Espresso (`v6.7`) and the AiiDA relaxation workflow [3]. The `fast` relaxation protocol without magnetization. For complete documentation of relaxation parameters see the Supplementary of [3]. Note for pseudo-potentials this protocol uses the `efficiency` set of the SSSP library `1.2.1` [4]. We use the PBE exchange-correlation functional for all relaxations.

## 1.2 Hyperparameters for model training

We document all hyperparameters used for training the SCN models in the accompanying file `hyperparams.yml`. This set was modified from the offical SCN implementation in the OCP repo [1]. In particular a much smaller model was used than the state-of-the-art SCN results.

## 1.3 Initializing SCF runs in Quantum Espresso with learned densities

An SCF run may be initialized with a custom density, e.g. one generated from a machine-learning model, using the `startingpot` input parameter of Quantum Espresso (QE) [2]. Importantly, we use QE compiled with `HDF5` support, rather than machine-dependent `dat` binary files. To initialize an SCF run with a learned density, we follow the file format of the `HDF5` charge density files expected by QE. This format includes coefficients of the *reciprocal* charge densities and their associated Miller indices. Only data with associated reciprocal space vector $G$ with $|G|^2 \leq$ `ecutrho` is written, where `ecutrho` is the energy cutoff for charge density in suitable units. We validate our charge-density read/write implementation is correct by checking that convergence of ground-truth densities are unaffected by reading/writing.