# OpenReview forum: "Towards Combinatorial Generalization for Catalysts: A Kohn-Sham Charge-Density Approach"
_NeurIPS.cc/2023/Conference — NeurIPS 2023 poster_

### Official Review · Reviewer_7uBr · 2023-06-25

**Soundness:** 3 good
**Presentation:** 3 good
**Contribution:** 3 good
**Rating:** 7
**Confidence:** 4

**Summary:**

The paper proposes a new learning framework for using machine learning to approximate chemical simulations. While many papers focus on direct property prediction (e.g. energy, forces) when applying machine learning to chemical simulation, this paper motivates the prediction of charge-density which holds promise for generalizable models across different materials systems. Following the introduction and motivation of the problem, the paper then describes related work with relevant methods in equivariant graph neural networks, materials property prediction and model generalization. Next, the authors describe their method for generating a charge-density based DFT dataset using an open-source computational chemistry engine and provide relevant statistics for their dataset, as well as a new metric (s-AUC) to measure modeling performance. The authors then describe the results of training a modestly sized SCN network on their dataset and provide an analysis of the SCN model's performance as part of the charge density prediction workflow compared to the currently applied method in Quantum Espresso calculations. Overall, the results indicate that the SCN can in some cases similar precision to the original method while being significantly more compute efficient.

**Strengths:**

The paper has the following strengths:
* Originality: The paper proposes a new learning target for computational chemistry simulations, charge-density, that is motivated by a deep understanding of the physics and chemistry of the Kohn-Sham equations. Based on authors' review, there is few prior works addressing this question even though proper models could lead to greater generalizability.
* Quality: The authors define a new performance metric (s-AUC), define a new dataset to ensure consistency and measure modeling error rates to a high precision.
* Clarity: The performance metrics are generally well defined and the plots in Figure 4 and Figure 5 provide a good overview of the SCN model's performance.
* Significance: Properly training charge-density models can be significant for the application of machine learning models to computational chemistry. The problem statement has good originality and significance.

**Weaknesses:**

The paper could be improved by:
* Providing greater summary statistics of the SCN model's performance across the entire dataset studied. While Figure 4 and Figure 5 are useful, it is currently hard to understand how well SCN performs across the entire dataset. Table 2 provides some statistics, but the Figure caption is not informative in trying to understand the content of the Table. [Clarity, Quality]
* It would be nice to see how effective charge-density calculation can lead to the prediction of properties found in other studies. The authors mention some reference in the text, but it would be nice to see some of those demonstrated in the paper and thereby showcase how the two methods compare. [Significance]
* The authors mention the release of related code and data in the supplementary material. Given that the dataset and workflow are the primary contributions of this work, it would be nice to see that clarified in the main paper. [Clarity]
* It would have been nice to see the evaluation of additional models on the generated dataset or at least an explanation as to why they were not considered. [Quality]
* Additional data related to compute efficiency increases of the proposed method would help make a stronger case for the potential benefits of the method. [Significance]

**Questions:**

Some questions for the authors:
* Can you provide additional detail on the types of materials systems your SCN model performed well on in addition to the binary and ternary classification? It could be interesting to look at generalization from the materials construction perspective.
* Can you provide additional details for the potential compute benefits of using SCN for charge density prediction?
* The high-precision learning motivation you have in Section 2.4 is quite relevant to this paper. Could you motivate this further in terms of high precision is needed. This might tie into applying these models in the computational workflow that you briefly mention, which would also be nice to expand upon.

**Limitations:**

The authors provide a discussion of limitations in their paper, which list a set of relevant points. It would be nice to include a discussion on other types of open-source solvers, i.e. not Quantum Espresso, as well.

---

> ### Author Rebuttal · Authors · 2023-08-09
>
> Thanks for your review.
>
> Regarding the performance of SCN on the entire dataset, please see the distributions of metrics in Figures 1 and 2. Most scores fall into this range of -0.5 and 3 in Figure 1, and between -0.2 and 0.8 in Figure 2.
>
> Regarding other models, we did try several other models like  Schnet, Dimenet++, and Gemnet in our preliminary experiments, but found SCN to perform the best. We are happy to include these other models in the next revision if it is a concern to you.
>
> We will certainly clarify our intent to release all code and data in the main text.
>
> (0)
>
> Regarding computing other properties from the charge density, in addition to the energy bias check, we also compared the band gaps computed from the ground-truth and ML densities. We do so using the built-in tool in ASE [2]. We find 95% of values to be exactly the same, with maximum relative error at $7.8\times 10^{-5}$.
>
> (1)
>
> Analyzing top performers for patterns is an interesting idea and could useful for improving the model or dataset. We attempted such an analysis from a view different angles, but didn't find anything conclusive to report. This is an interesting question for future work.
>
>
> (2)
>
> Yes, we have now included percentage of iterations saved. See Table 2 and Figure 2.
>
> (3)
>
> We are glad the reviewer found the section on high-precision relevant. We included it because the precision requirements for DFT applications are often quite high, e.g. the convergence threshold we used was 1e-10.
>
> We want to point out the distinction between high-precision in training and testing. We think the later, concerning generalization to new structures with high precision, is a particularly challenging.
>
> Although our work doesn't contribute any specific training techniques as in [0], we are able to achieve high-precision by interfacing with the SCF cycle in a natural way: through initialization with learned densities. This "hybrid approach" is echoed in a number of places, e.g. [1], and we believe is an increasingly important alternative to approaches purely with ML.
>
> We will add this discussion in the next revision.
>
>
> * [0] https://arxiv.org/abs/2210.13447
> * [1] https://www.nature.com/articles/s41467-019-12875-2
> * [2] https://wiki.fysik.dtu.dk/ase/ase/dft/bandgap.html

---

> > ### Comment · Reviewer_7uBr · 2023-08-14
> > **Thank for the additional details**
> >
> > The authors have addressed most of my concerns. I am adjusting my score accordingly.

---

### Official Review · Reviewer_XGHX · 2023-07-03

**Soundness:** 2 fair
**Presentation:** 2 fair
**Contribution:** 3 good
**Rating:** 5
**Confidence:** 2

**Summary:**

In this work, the researchers created a novel dataset of bulk catalyst relaxations with charge densities using open-source software. They aimed to investigate the phenomenon of combinatorial generalization, which refers to the ability of a model to effectively perform on new combinations of elements that were not seen during training. To evaluate the performance of their model, the researchers proposed a metric based on the comparison of convergence rates between their approach and standard charge-density initialization baselines. This metric served as an indicator of the quality of the results obtained. The experimental results demonstrated evidence of combinatorial generalization, indicating that the model performed well on new combinations of elements that were unseen during training. The proposed metric showed that the learned charge densities produced better results than the standard baseline in over 50% of the cases.

**Strengths:**

The strengths of this paper are:

* Contribution done in the field of catalysis.
* Compute a new medium-scale DFT dataset of relaxed bulk catalyst structures with charge densities.
* Proposal of a new metric used to verify the quality of results.

**Weaknesses:**

* The reference to figures is confusing. In lines 195 and 196 you write " ...Figures 4.2 and  4.2 respectively.". Same in lines 210 and 211.
* There is no discussion regarding Figures 4 and 5 and the results they represent.
* Although the code is not provided there is no high-level explanation of the architecture of the model.

**Questions:**

* What is the loss used for the proposed method?
* Are the results reproducible?
* Can the new proposed dataset be used by different approaches or is specific to the proposed method?

**Limitations:**

While the paper presents a novel approach and introduces a new dataset, one limitation is the absence of the dataset itself and the code required to replicate the results and gain a deeper understanding of the proposed method. The availability of the dataset and code is crucial for reproducibility, as it allows us to verify the findings. Therefore, it is essential to consider releasing the dataset and code to facilitate transparency and reproducibility.

---

> ### Author Rebuttal · Authors · 2023-08-09
>
> Thanks for your review. We realize we didn't make a number of details clear in the main text. We will make sure to add these in the next revision.
>
> (1)
>
> The model is trained with an regression loss. Following other work in ML for catalysis, final prediction losses are reported as "mean absolute error" (MAE) [0].
>
> (2)
>
> We agree with the reviewer on the importance of reproducibility and fully intend to release all code and data upon publication. We realize we didn't make this clear in the main text, only the Supplementary. We also describe our code and hyperparams there, which should ensure full reproducibility of our results.
>
> (3)
>
> We created our dataset to specifically study combinatorial generalization in catalyst systems. However our experience tells us that researchers often find new and creative ways to use datasets that go beyond the original design. For instance, as another reviewer pointed out, other learning targets such as planewaves or atomic orbitals are alternative approaches. These would be possible with our dataset.
>
> [0] https://arxiv.org/abs/2206.14331

---

> > ### Comment · Reviewer_XGHX · 2023-08-20
> >
> > I want to thank the authors for addressing my inquiries. I will adjust my score accordingly.

---

### Official Review · Reviewer_Fg4b · 2023-07-06

**Soundness:** 3 good
**Presentation:** 3 good
**Contribution:** 2 fair
**Rating:** 5
**Confidence:** 3

**Summary:**

- The authors investigate charge density modeling in catalyst systems. They generated a dataset using Quantum Espresso on Materials Project, defined a metric of SCF convergence savings, and tested models, focusing on generalization and comparing it to ACP initialization. They used GNNs with virtual atoms as query points. With reduced size SCN, 45% of binary cases and 63% of ternary cases achieved positive savings.

**Strengths:**

- This tackles an important problem and is a good complement to all the models directly predicting properties like energies and forces
- The new dataset will be useful in this field
- The paper is clearly written and easy to follow

**Weaknesses:**

- Only 50% of samples achieving positive savings, and binaries doing worse, do not seem like great results or something usable in practice.
- More comprehensive analysis would be nice, e.g. why do errors occur near core regions of atoms (and is that good or bad?)
- This paper only tests one model, which is based off an existing one, limiting the scope of contribution. Could there be different architectures better suited for charge density prediction?

**Questions:**

- How does performance compare between 1. exact materials (stoichiometry + crystal structure) in train, 2. elemental combinations that were seen in train (but not the same material), and 3. elements that were seen in train but combinations that were not? (Assuming you can't test on individual elements that were not seen in train). I am curious how high performance you can even get on in-distribution materials, and whether the low generalization performance is due to lack of generalization of just underfitting.

- How granular is your grid? How granular does it need to be to be useful?
- Is it possible to make it faster but wrong? i.e. how does one make sure that the converged result is still correct?


**Limitations:**

- The authors acknowledge several limitations as potential future work: larger of systems in dataset, larger models, catalyst systems with adsorbates in addition to bulks alone, understanding why binaries are worse than ternaries.

---

> ### Author Rebuttal · Authors · 2023-08-09
>
> Thanks for your review.
>
> (0)
>
> We did try other model architectures such as Schnet, Dimenet++, and Gemnet, but we found the SCN to be the best performing. Our contribution is not specifically about finding the best architecture for density prediction, rather it is about investigating a specific kind of generalization, which we believe is interesting in its own right. However we are happy to include results with other models in the next revision.
>
> (1)
>
> Regarding underfitting, we find performance on the training set is generally better than on the test set, e.g. see MAE and savings results for the train split in Figures 1,2 and 3 in the attached. Our model was trained for 420k iterations (batchsize=44). It's possible that further training would help. But our experience tells us that adding more data may be more important.
>
> It is true in this work that we do not address variation that occurs *within* a combination (i.e. item 2 from question 1). This kind of variation is certainly important, but it is not the focus of our work.
>
>
> (2)
>
> Grid sizes vary by structure. For instance, an example of a small grid is mp-129 with size 19x19x19 (= 6859) and a large one is mp-23329 with 127x127x127 (= 2048383).
>
> In plane-wave DFT, grid size is determined by primarily by the the energy cutoffs and cell size. We use the energy cutoffs recommended by the relaxation workflows from [1,2], which are the result of a rigorous study of pseudopotentials.
>
> (3)
>
> This is a good question. We checked for differences between the energies from converged ML and ground-truth densities and found no bias. Empirically this supports the claim that ML densities are converging to the "correct" solution.
>
> * [1] https://www.nature.com/articles/s41524-021-00594-6
> * [2] https://www.nature.com/articles/s41524-018-0127-2

---

> > ### Comment · Reviewer_Fg4b · 2023-08-18
> >
> > Thank you for the response. The improved performance looks promising, and it's good to see confirmation that the solutions are still valid. I have increased my score.
> >
> > > it is about investigating a specific kind of generalization, which we believe is interesting in its own right
> >
> > Generalization is interesting, but I don't think the messaging should emphasize it that much as a contribution because generalization is naturally important in all ML applications. Also, if generalization is emphasized, I would expect more in depth analysis on in-domain vs. out-of-domain performance, or different kinds of generalization, or more error analysis.

---

> > > ### Author Response · Authors · 2023-08-18
> > >
> > > We appreciate the reviewer's concern about our messaging around generalization.
> > >
> > > We agree that generalization is naturally important in ML, and that there are aspects of it that we don't address.
> > >
> > > We will be careful to keep our claims narrowed to a specific kind of out-of-distribution generalization, involving new combinations of elements, for which we have evidence.

---

### Official Review · Reviewer_iUWH · 2023-07-19

**Soundness:** 3 good
**Presentation:** 4 excellent
**Contribution:** 2 fair
**Rating:** 5
**Confidence:** 5

**Summary:**

The authors build and train approximately E(3) equivariant graph neural networks based on Spherical Channel Network to learn the Kohn-Sham Charge-Density scalar field as a means of accelerating density functional theory (DFT) calculations by shortening the number of self-consistent field calculations needed to achieve a converged density. The authors are motivated by finding a generalizable approach to evaluate the energetics of the combinatorial space of binary, ternary, and beyond heterogeneous catalytic surfaces, which the authors point out is poorly predicted by current pure ML approaches. The model uses a virtual node to sample the value of the charge density at specific locations that are compared to ground truth DFT calculations computed at the level of PBE using the open-source package Quantum Espresso.



**Strengths:**

The paper is extremely clear and well written, especially considering the need to present internal details of DFT to a general ML audience. The authors emphasize that combinatorial generalization, the ability of a model to generalize to catalytic surfaces that have combinatorial many possible substitution of constituent atoms, is a necessary metric to satisfy if ML will have impact in the heterogeneous catalysis space.

**Weaknesses:**

The largest weaknesses of the paper are that the charge density predictions are made on a relatively few and specific set of structures (it is unclear how well this approach will generalize to more diverse structures and atom type distributions) and that results are not presented in a way makes it easy to assess the impact of the results on the broader ML+DFT community.

For this reason, I give a “weak reject” but very much hope the authors will continue to pursue this line of work as I believe it to be a powerful approach to aiding materials design. I would also consider changing my rating if the authors can quantify the benefits of their approach more strongly. I give more details below in the Questions section.

**Questions:**

Below I give concrete feedback on specific parts of the paper section by section.

> Introduction
Very clear and well written. A concise and thorough description of the problem statement and goals of the paper.

> Related Work
Another reference that could be included in related work is the following, which explicitly discusses using ML predicted charge densities of molecules as starting points for calculations:
https://www.nature.com/articles/s41467-019-12875-2
In that paper they found minimal advantage to using ML charge densities but they only explored this for a small number of molecules (possibly only 1). Do your results align or disagree with their findings?

> Methods
>> 3.1
I would be helpful to discuss further how the query points are selected. Additionally, it would be helpful to discuss the pros and cons of using a query point vs predicting basis functions (either plane waves or linear combination of atomic orbitals). I can appreciate that the query point is most flexible but you do seem to need a very large number of samples per structure. Quantitative comparison of the different charge density representations would be helpful.

>> 3.2
One concern I have about using small unit cells is that this significantly limits what “patterns'' of different atomic neighborhoods the model sees and limits the significance of your results – since we don’t get a sense for how much data and samples would be needed to get comparable performance across more diverse environments (even if restricted to the same lattice). There are many ways to choose sites of a lattice but if the unit cells have lattices vectors only ~4 Å long, you’re only going to be able to explore fairly high symmetry configurations.

Regarding the data selection based on oxidation state, one could consider encoding the oxidation state as an input feature in such a way that the model may be able to learn to generalize. For example, one could augment each atom type with an oxidation state feature.

>> 3.3
Typo: “this baseline is high non-trivial”

> Results
The labels of Figures 2 and 3 are very small. Please improve the label sizes, choose an appropriate number of significant figures reported for MAE and Max Error, and specify which materials (mp-id(?) Or chemical formula) are being shown.

>> 4.2
Type: “of of”

>> 4.3
How are these predictions converted to your input charge density to Quantum Espresso? My understanding is that you need to convert it into plane waves – are there any subtleties in doing this?

The paper would be improved if the authors gave more specific metrics on the impact their results imply for performing general DFT calculations. The authors could better quantize how much compute is used for DFT calculations and how much better initial charge density predictions could save in compute. This would be a more powerful metric than just the number of examples that see any improvement in SCF steps. For example, a large fraction of the USA’s National Laboratory supercomputing resources are spent computing DFT – how much would better starting points save for a given level of accuracy? How much accuracy is needed for ML charge density prediction to be practical for these facilities to adopt into workflows?

Something that might seem like a silly benchmark but may be informative is what is the likelihood of reducing the number of SCF steps with random noise on the charge density (physically motivated noise but still noise). This would give more information as to whether what you are predicting requires learning or whether it’s due to the finicky nature of SCF calculations.

An additional sanity check that I would like to see is what is the difference between the converged charge density and the energy for the standard vs. ML accelerated calculations. It’s important to check whether the introduction of an ML derived initial charge density introduces any bias in the calculation results. It may very well be that there is no difference – but it’s important to check!

> Broader Impact
Here’s another great place to discuss how many hundreds of millions of hours of compute are spent on DFT by researchers and what potential savings good charge density starting points might enable.


**Limitations:**

Yes, the authors are very clear about the limitations of their methods and what can be improved on. They are also clear about what are open questions in the field (section 2.1) and what is and is not known. The authors come off as clear, knowledgeable, and thoughtful.

The authors are aiming to accelerate DFT calculations which are already used used broadly. They specifically address the potential concerns of making these insights more readily available.

---

> ### Author Rebuttal · Authors · 2023-08-09
>
> We thank the reviewer for the thoughtful review.
>
> (1)
>
> We are happy to add a discussion of Schütt et al. (2019) to our related work section. In short we think our results are complementary. The primary differences between our works is that (1) we study learning of the charge density for bulk catalyst systems using the SCN model with an explicit focus on combinatorial generalization (2) they study learning elements of the Hamiltonian matrix directly for single molecules (water, malonaldehyde, uracil) using the SchnetOrb model (a modified version of Schnet, an early predecessor to SCN). They show SCF savings for malonaldehype (Fig 4).
>
> (2)
> We select query points (QP) from the (real-space) grid associated to each structure. We take a maximal approach and assign a query point to  *all*  points in the grid. The possibility of more efficient selection method, e.g. through sampling, is an interesting one. In early experiments we tried some sampling approaches, but found this sometimes lead to prediction errors in poorly sampled regions, which we found detrimental to SCF convergence. We leave further investigation of this for future work. We note that this and other optimizations do not change the main message of our paper, which is evidence for combinatorial generalization in catalyst systems.
>
> Regarding a QP approach vs predicting basis functions, we choose the former because it naturally fits into the existing framework of geometric graph learning approaches, e.g. the QP is simply another geometric node in the graph and thus any equivariant properties of the model are not violated. We had considered learning the density in reciprocal space, but it is not clear to us how to define graphs there.
>
> (3)
> We choose to limit cell sizes for reasons of computational burden, in particular to help constrain the maximum grid size per structure. For example, one large grid we found was ~10M points, which would occupy a substantial fraction of our dataset and therefore our training budget. To emphasize more diversity in structures, in the sense of different combinations, we imposed this constraint.
>
> Additionally, for the new larger dataset, we relaxed these constraints to (1) max 12 atoms per cell (2) maximum volume of 200 A^3, and see improved results. Generalization to yet larger and more diverse structures still remains to be demonstrated, and is an interesting point of continuation for future work.
>
> (4)
>
> Thanks for the suggestion about encoding oxidation states as a feature. One tricky part about this is how to do it in such a way to that preserves equivariance. One approach we tried was to encode oxidation states using the atomic index (i.e. every unique oxidation state gets a unique index). But we found this to not improve results. We leave this for future work.
>
> (5)
>
> We appreciate the reviewer's concern about presenting our results in a way to better assess impact on the ML+DFT community.
>
> We now include a measure of the relative number of SCF iterations saved at convergence. See Table 2 and Figure 2 in the attached.
>
> We agree this work may potentially have large impact through saving costs, e.g. at national labs. Thanks for bringing that point up. We think our work is a step in that direction.
>
> We weren't able to perform the "physically motivated noise" benchmark you proposed, but generally our experience is the SCF cycle is very sensitive to noise. Let us know in the discussion phase if this is still a concern for you.
>
> Thanks for bringing up the check energy bias. We checked this and found virtually no difference between the energies. See the general comments for details.
>
> (6)
>
> Regarding converting predictions to input charge density for Quantum Espresso, we give some details on this in the Supplementary. In short all that is required is to match the file format of charge density HDF5 files which QE natively supports. We did find some sensitivity regarding the order in which data is written, but we were able to verify our implementation on ground-truth cases.

---

> > ### Comment · Reviewer_iUWH · 2023-08-17
> > **Updated score.**
> >
> > I have updated my score in response to the authors rebuttal.

---

### Author Rebuttal · Authors · 2023-08-09

We thank the reviewers for their feedback. We address common concerns of reviewers below:

* (1) strength of performance and number of structures
* (2) assessing impact of results on ML+DFT community
* (3) comprehensiveness of evaluation
* (4) bias of learned densities

---

(1)

To address concerns about performance, we have scaled up our dataset by a factor of 8.5x more structures. We now train on 47 unary and 392 binary catalysts, spanning 73M samples of (graph, density) pairs.  We substantially increase the test set, now evaluating on 360 binaries and 1116 ternaries unseen at train time, spanning over 450M samples. We follow the same data generation procedure as described in the main text, except for relaxing constraints on the cell size (now 200 Ang^3) and number of atoms (now 12). All told, our dataset now spans 1915 structures and 525M samples.

Performance on this new data is much improved. In terms of the s-AUC metric, we improve upon the baseline in 83% and 86% of binary and ternary test cases respectively. We summarize these new results in Table 1.

(2)

We care about presenting our results in a way that makes it easier to assess impact on the ML+DFT community. To that end, we have also computed savings in terms of the percentage of SCF iterations saved at convergence. We think this should be more interpretable to the community. We find that we save on average of 13% of iterations in both binary and ternary cases. We report summary statistics for iteration savings  in Table 2.

To put it as suggested by one reviewer, supposing that the USA National Labs spend \\$100M on DFT per year and negligible cost of inference, then a 13% reduction in SCF iterations would yield a savings of \\$13M. There are of course a number of practical details to work out to achieve this, but we believe this highlights the potential impact of our work.

(3)

We care about providing the full picture of our results. To that end, we show the distributions over structures of s-AUC scores and iterations saved in Figures 1 and 2. These plots and the summary statistics give the complete picture.

(4)

Some reviewers raised the question of bias, i.e. whether learned densities might converge to a different solution than the ground truth. To check this, we compared the energies at convergence between the ground-truth and learned cases. We find almost zero difference between these energy values. On the test set we found 88% of structures had exactly zero difference in energy values. Of the cases with non-zero error, the relative error on average was $1.2\times 10^{-11}$ and at a maximum of $1.5 \times 10^{-9}$. Empirically it appears there is very little difference.

---

### Decision · Program_Chairs · 2023-09-21

**Decision:**

Accept (poster)

**Comment:**

This paper tackles the problem of charge density modeling in catalyst systems, an important one in the area of ML + Materials Science.  On the one hand, there was unanimity among reviewers that the paper tackles an important problem, that it is presented clearly and accessibly to a ML audience, and that it is likely to be of significance to the community. In addition, the reviewers appreciated the usefulness of the dataset introduced in this paper. On the other hand, there were concerns about the scope of the experimental validation (predictions made on a narrow set of structures, only a single model tested, etc). The rebuttal by the authors, however, was thorough and addressed many of these concerns. Overall, despite its limitations, this paper is likely to be of interest to the ML + Density Functional Theory (DFT) community.